# Lymphotropic Viruses: Chronic Inflammation and Induction of Cancers

**DOI:** 10.3390/biology9110390

**Published:** 2020-11-10

**Authors:** Edward W. Harhaj, Noula Shembade

**Affiliations:** 1Department of Microbiology and Immunology, Penn State College of Medicine, Hershey, PA 17033, USA; ewh110@psu.edu; 2Department of Microbiology and Immunology, Sylvester Comprehensive Cancer Center Miller School of Medicine, University of Miami, Miami, FL 33136, USA

**Keywords:** HTLV-1, EBV, EBV, STAT, NF-κB, cell surface molecules (CSMs), inflammation, cancers

## Abstract

**Simple Summary:**

Infection with viruses such as HTLV-1, EBV and KSHV has been linked to many cancers, including leukemias and lymphomas in humans worldwide. These viruses establish life-long latency after initial infections. Currently, there are no effective treatments for the prevention of cancers associated with these viruses. Studies have shown that chronic inflammation mediated by the transcription factors STAT3 and NF-κB in HTLV-1, EBV and KSHV-infected cells play critical roles in the development of viral-associated cancers. Inhibition of STAT3 and NF-κB activation in cells harboring these latent viruses leads to their destruction and the production of new virus particles. De novo infection by these viruses induces rapid NF-κB and STAT3 activation and creates a favorable environment for virus entry into host cells and viral latency. However, the host factors and the mechanisms required for rapid NF-κB and STAT3 activation during de novo infection and latency by these viruses are largely unknown. In this review, we will discuss the mechanisms that are specifically involved in NF-κB and STAT3 activation during de novo infection and latency by KSHV, EBV and HTLV-1.

**Abstract:**

Inflammation induced by transcription factors, including Signal Transducers and Activators of Transcription (STATs) and NF-κB, in response to microbial pathogenic infections and ligand dependent receptors stimulation are critical for controlling infections. However, uncontrolled inflammation induced by these transcription factors could lead to immune dysfunction, persistent infection, inflammatory related diseases and the development of cancers. Although the induction of innate immunity and inflammation in response to viral infection is important to control virus replication, its effects can be modulated by lymphotropic viruses including human T-cell leukemia virus type 1 (HTLV-1), Κaposi’s sarcoma herpesvirus (KSHV), and Epstein Barr virus (EBV) during de novo infection as well as latent infection. These lymphotropic viruses persistently activate JAK-STAT and NF-κB pathways. Long-term STAT and NF-κB activation by these viruses leads to the induction of chronic inflammation, which can support the persistence of these viruses and promote virus-mediated cancers. Here, we review how HTLV-1, KSHV and EBV hijack the function of host cell surface molecules (CSMs), which are involved in the regulation of chronic inflammation, innate and adaptive immune responses, cell death and the restoration of tissue homeostasis. Thus, better understanding of CSMs-mediated chronic activation of STATs and NF-κB pathways in lymphotropic virus-infected cells may pave the way for therapeutic intervention in malignancies caused by lymphotropic viruses.

## 1. Introduction

### Lymphotropic Viruses and Cancers

Infection with lymphotropic viruses, including Epstein Barr virus/human herpesvirus 4 (EBV/HH4), Κaposi’s sarcoma herpesvirus/human herpesvirus 8 (KSHV/HHV8), and human T-cell leukemia virus type 1 (HTLV-1), is associated with the development of cancers worldwide [1]. The rapid induction of inflammation by these viruses plays a critical role in the development of many cancers [2,3]. While both EBV and KSHV are linear double stranded DNA viruses [4,5], HTLV-1 is a retrovirus [6]. EBV was the first human lymphotropic virus associated with B-cell and T-cell lymphoproliferative disease, discovered in 1964 [7,8,9], and one of the most common viral infections in humans [10,11]. EBV has been associated with the etiology of multiple human cancers of both lymphoid and epithelial origin, including Burkitt’s Lymphoma, Hodgkin lymphoma, and Nasopharyngeal Carcinoma and EBV-associated gastric cancer [12]. KSHV, discovered in 1994, is associated with Kaposi’s sarcoma (KS), and two lymphoproliferative disorders called Primary effusion lymphoma (PEL) and multicentric Castleman’s Disease (MCD) [13,14,15]. HTLV-1 was discovered in 1980 and is linked to two main diseases: a malignant T-cell lymphoproliferation known as Adult T-cell Leukemia/Lymphoma (ATLL) and Tropical Spastic Paraparesis/HTLV-1-Associated Myelopathy (TSP/HAM) [16,17,18]. EBV, KSHV and HTLV-1 often establish a life-long asymptomatic latent infection to develop these cancers [19]. These tumor viruses exhibit a tropism for either B-, T-, epithelial or endothelial cells [19,20,21].

## 2. Inflammation and Oncogenesis

### 2.1. NF-κB

Nuclear factor-κB (NF-κB), a family of transcription factors, is ubiquitously expressed and is critical for the regulation of innate and adaptive immune functions [22,23]. NF-κB family members consist of NF-κB1 (p50), NF-κB2 (p52), RelA (p65), RelB and c-Rel, which can form various combinations of homo- and heterodimers [22,23]. All NF-κB subunits contain a conserved Rel homology domain that facilitates DNA binding, nuclear localization and dimerization of NF-κB proteins. NF-κB is activated by canonical and non-canonical signaling pathways. Canonical NF-κB signaling is responsible for the induction of pro-inflammatory cytokines, chemokines and inflammatory mediators in different types of innate immune cells [24]. The non-canonical NF-κB signaling pathway plays critical roles in lymphoid organogenesis, B-cell survival and maturation, dendritic cell activation and bone metabolism [25,26]. The signaling components and receptors involved in the activation of both canonical and non-canonical pathways are distinct.

The NF-κB proteins are normally held inactive in the cytoplasm by the inhibitor IκBα and other IκB family members, which contain ankyrin repeat domains. The NF-κB1 and NF-κB2 proteins are derived from precursor proteins, p105 and p100, respectively. In response to receptor stimulation or microbial infection, IκBα is rapidly phosphorylated by the IκB kinase (IKK) complex, consisting of two catalytically active kinases, IKKα and IKKβ, and a regulatory subunit IKKγ (NEMO), in the canonical NF-κB signaling pathway (Figure 1) [27]. Phosphorylation of IκBα leads to its polyubiquitination and proteasome-dependent degradation, which allows bound NF-κB dimers to translocate to the nucleus and activate gene expression (Figure 1). In contrast to the canonical NF-κB pathway, the non-canonical NF-κB pathway is activated by TNF receptor superfamily members, including BAFFR, CD40, lymphotoxin β receptor and RANK. The non-canonical NF-κB activation requires kinases NIK and IKKα that mediate p100 phosphorylation, ubiquitination, and processing to p52, which results in the generation of RelB and p52 dimers.

Chronic activation of NF-κB leads to the development of various autoimmune, inflammatory-related disorders and solid tumors, as well as leukemia and lymphoma. The transient transcriptional activity of NF-κB is normally maintained by one of the NF-κB target genes, IκBα, in a negative feedback loop, after stimulation with NF-κB activating receptors. Furthermore, a number of other NF-κB negative regulatory molecules, including A20 and Cyld, are involved in maintaining transient activation of NF-κB to prevent autoimmune, inflammatory related disorders and cancers [28,29,30].

### 2.2. STAT3

STAT3 is a member of the STAT (Signal Transducers and Activators of Transcription) transcription factor family discovered in 1994 [31,32,33]. The transcriptional activation of STAT3 by phosphorylation at tyrosine 705 (Y705) is regulated by a diverse array of cytokine receptors (e.g., Interleukin (IL)-6, IL-11, IL-23), growth factor receptors (e.g., epidermal growth factor (EGF)), G protein-coupled receptors (GPCRs) (e.g., sphingosine-1-phosphate receptor-1 (S1PR1), Angiotensin II type I receptor (AT1R)), ultraviolet (UV) radiation, oncogenes and stress. Although transient STAT3 phosphorylation and activation is essential for the development of immune cells such as T cells and B cells, cell growth and proliferation, cell migration, and cell death, chronic and hyper-activated STAT3 results in cancer-promoting inflammation, increased cell proliferation, cell survival, angiogenesis and metastases, and dysregulation of anti-tumor immunity [34,35]. Furthermore, activated STAT3 is a critical regulator of cell energy metabolism, which could have a significant impact on tumor transformation and growth [36]. Many of the STAT3-induced genes are cytokines and growth factors, which again bind to their receptors and activate STAT3 thereby maintaining chronic STAT3 activation in many cancers [37]. 

In an unstimulated cell, STAT3 is in an inactive state in the cytoplasm; however, it is activated by direct phosphorylation at tyrosine 705 and serine 727 residues by the receptor-associated Janus kinases (JAKs), tyrosine kinases, and select non-receptor tyrosine kinases such as EGFR, SRC and AB [34,35,38]. Phosphorylated STAT3 dimerizes or oligomerizes prior to nuclear translocation, DNA binding and induction of its target genes (Figure 1) [35,39]. Transient activation of STAT3 is a critical regulatory mechanism and is mediated by a variety of post-translational modifications of STAT3 and direct interactions with negative regulatory proteins [40]. Biologically active STAT3 dimers are destabilized by several tyrosine phosphatases, including PTPN6, PTPRD, PTPRT and PTPN11 [41]. Inactivation of these phosphatases by either genetic mutations or epigenetic regulation leads to chronic activation of STAT3 phosphorylation in tumor cells [42,43,44,45,46]. Most importantly, STAT3 activation is negatively regulated at the receptor level by STAT3 target genes, suppressors of cytokine signaling (SOCS) E3 ubiquitin ligases, which induce the degradation of cytokine receptor complexes [47]. STAT3 activation is also negatively regulated by protein inhibitor of activated STAT (PIAS), an E3 SUMO-protein ligase in the nucleolus (Figure 1) [48,49]. 

### 2.3. Crosstalk between NF-κB and STAT3 in Cancers

Recent studies have found that chronic infection or inflammation mediated by NF-κB and STAT3 plays a pivotal role in linking inflammation and cancer [50,51,52]. The genes induced by both NF-κB and STAT3 are critical regulators of cellular proliferation, angiogenesis, genomic instability, resistance to apoptosis, invasion and metastasis of tumor cells [53]. NF-κB and STAT3 activation are highly connected, and interact functionally with each other in a number of layers. Activation of NF-κB induces several cytokines and growth factors, such as IL-6, which is a major STAT3 activator. STAT3 activation also contributes to NF-κB activation by binding to RelA/p65 in the nucleus and enhancing RelA/p65 reversible acetylation, thereby prolonging its nuclear retention [54,55]. In addition, NF-κB and STAT3 bind together to a subset of gene promoters to cooperatively induce expression of their target genes [56].

## 3. Role of Cell Surface Molecules (CSMs) in EBV, KSHV or HTLV-1-Induced Inflammation during De Novo Infection

Most enveloped viruses, including EBV, KSHV and HTLV-1, rely on an optimal amount of certain cell surface molecules (CSMs) expressed on host cells for attachment, cell entry and spread. Inflammation induced by NF-κB and STAT3 upon attachment of KSHV or EBV envelope proteins to CSMs is necessary for their entry into target cells, for the establishment of latency and for the viral life cycle. However, it is unclear if HTLV-1 attachment to CSMs induces NF-κB and STAT3 activation and inflammation. In addition, the mechanisms and host CSMs involved in promoting the entry and inflammation of these viruses are not well understood. 

### 3.1. CSMs in EBV-Induced Inflammation during De Novo Infection

EBV envelope glycoprotein 350 (gp350) binds to the cell surface receptor CD21 (type II complement receptor or CR2), which forms membrane complexes with CD35/CR1 and CD19, expressed on many human cell types such as B and T-lymphocytes, thymocytes, epithelial cells, follicular dendritic cells and endothelial cells. This membrane complex initiates EBV entry into the target cells [57,58,59]. At the same time, binding of EBV envelope gp350 to these receptor complexes rapidly induces activation of IKK and NF-κB (Figure 2) [60,61,62,63]. This activation of IKK and NF-κB is primarily mediated by phosphatidylinositol-3-kinase (PI3K) and protein kinase C (PKC) during EBV de novo infection. Similarly, STAT3 phosphorylation at Y705 and activation by EBV de novo infection has been reported [64,65]. Although STAT3 activation by EBV de novo infection is mediated through JAKs, the mechanism and CSMs involved in activating JAKs upon EBV de novo infection are not well understood. The activation of NF-κB and STAT3 is critical for the cell survival, proliferation, and adhesion of EBV infected cells, as well as immortalization [65,66,67].

### 3.2. CSMs in KSHV-Induced Inflammation during De Novo Infection

Like EBV, KSHV envelope glycoprotein gB, K8.1, and the complement control protein (KCP) promote attachment to cell surface associated heparan sulfate (HS), integrins (α3β1, αVβ3, and αVβ5) and EphA2 receptor tyrosine kinase (EphA2R), which initiates KSHV DNA entry into target cells [68]. Following attachment of KSHV glycoproteins to cell surface receptors, key signaling molecules including Focal Adhesion Kinase (FAK), non-receptor cytoplasmic tyrosine kinase Src, and PI3K are activated, which results in the activation of NF-κB [69]. Binding of KSHV to its receptors has been shown to activate NF-κB by a mechanism involving IKKβ and IKKε (Figure 2) [70]. Our recent findings suggest that cell surface expressed Cell adhesion molecule 1 (CADM1) is critical for KSHV de novo infection-induced NF-κB activation [71]. Although CADM1 and IKKβ and IKKε kinases are critical for NF-κB activation upon KSHV de novo infection, the mechanisms of CADM1 and IKKβ and IKKε activation are unclear. In addition, if CADM1 is required for IKKβ and IKKε early activation during KSHV de novo infection is unknown. It has been shown that KSHV de novo infection rapidly induces STAT3 phosphorylation at Y705 [65]. Activation of STAT3 is mediated by JAK2 upon KSHV de novo infection (Figure 2). In addition to activation of JAK2 and STAT3 by KSHV de novo infection, the JAK2 and STAT3 activating cell surface receptor gp130 is induced by KSHV infection. This upregulation of receptor gp130 maintains constitutive phosphorylation of JAK2/STAT3 in KSHV-infected cells (Figure 2). KSHV de novo infection-induced NF-κB and STAT3 activation ensures cell growth, cell survival and KSHV latency.

### 3.3. CSMs in HTLV-1-Induced Inflammation during De Novo Infection

Unlike EBV and KSHV, cell free HTLV-1 virus has extremely low infectivity [72]. HTLV-1 infection occurs through cell-to-cell contact between infected donor cells and uninfected target cells, in which viral particles are transmitted via viral biofilms, cellular conduits or viral synapse formation with essential contributions from cellular surface expressing lymphocyte function-associated antigen-1 (LFA1) and its ligand intracellular adhesion receptor 1 (ICAM-1) [73,74,75,76,77]. Upon binding of HTLV-1 to its cell surface receptors, glucose transporter 1 (Glut-1), neuropilin-1 (NRP-1, BDCA-4), and heparan sulfate proteoglycans (HSPGs), the viral RNA is delivered into the cytoplasm, reverse transcribed, and integrated into the host genome to form a provirus. Although the main target cells of HTLV-1 in vivo are CD4+ T-cells, HTLV-1 can infect a wide variety of cells *in vitro*, including CD8+ T-cells, endothelial cells, B-lymphocytes, myeloid cells and fibroblasts, possibly due to ubiquitous expression of the receptor Glut-1 [78,79]. Although it is not well understood if NF-κB and STAT3 are activated during HTLV-1 de novo infection, we speculate that NF-κB and STAT3 are activated, as NF-κB and STAT3-regulated inflammatory molecules, including IFN-γ, IL-1, TNF-α, IL-6 and CXCL10, are produced during HTLV-1 de novo infection [80]. These inflammatory molecules produced during HTLV-1 de novo infection play critical roles in the immortalization, survival and proliferation of infected cells. Furthermore, many NF-κB and STAT3 target genes induced during HTLV-1 de novo infection are critical for HTLV-1-mediated dysregulation of host type I Interferon antiviral signaling [81,82].

## 4. Role of CSMs in the Induction of Inflammation during EBV, KSHV, and HTLV-1 Latency

Humans that have been infected with EBV, KSHV or HTLV-1 remain infected for life due to viral latency in B cells and T cells. Upon entry into the target cells of these viruses, the viral latent proteins, such as homologue of FADD-like interleukin (IL)-1β-converting enzyme (FLICE/caspase-8) inhibitory protein (vFLIP), latency-associated nuclear antigen (LANA), Viral interleukin-6 (vIL-6) of KSHV, EBV-encoded nuclear antigens (EBNA), latent membrane proteins (LMPs) of EBV, and the trans-activator protein, Tax, and HTLV-1 basic leucine zipper factor (HBZ) of HTLV-1 are expressed. Expression of these viral oncogenes maintains chronic activation of both NF-κB and STAT3, and inflammation required for the immortalization and transformation of EBV, KSHV and HTLV-1-infected cells [65,83]. 

### 4.1. CSMs and Chronic Inflammation during EBV Latency

EBV infection-induced CSMs such as CD40 and CD137 expression on EBV-infected T or NK cells play critical roles in the constitutive activation of NF-κB and survival of infected cells [84,85,86]. Studies have shown that chronic activation of STAT3 is thought to be triggered by IL-6, which is induced by persistent activation of NF-κB [87,88]. Activation of NF-κB and STAT3 following B lymphocyte infection with EBV in vitro leads to the induction of chronic inflammation, which is critical for the continuous proliferation of infected B cells and the establishment of lymphoblastoid cell lines (LCLs) [89,90]. During this latency, known as latency type III, a total of nine genes are expressed, including EBV-encoded RNAs (EBER 1 and 2), EBV nuclear antigens (EBNAs 1, 2, 3A, 3B, 3C, and LP) and EBV latent membrane proteins (LMPs 1, 2A, and 2B). These genes play a key role in cell proliferation, migration and survival [12,91]. During latency II, EBERs, EBNA-1, LMP1 and LMP2 are expressed and this is typically observed in EBV-associated Hodgkin’s lymphoma, NPC and T/NK cell lymphomas [92,93,94]. During latency I, only EBERs and EBNA-1 are expressed and this occurs in Burkitt’s lymphoma [95,96,97]. In addition to the latent infection, the lytic form of EBV infection also occurs occasionally to produce new virus particles. During EBV lytic infection BILF1, a viral G protein–coupled receptor (vGPCR) is expressed, which downregulates cell surface HLA class I expression and inhibits T cell recognition of EBV-infected cells [98,99,100].

Both LMP1 and BILF1 viral proteins are expressed on the cell surface and are persistently activated [66]. In addition to STAT3 activation by LMP1 via NF-κB-IL-6, LMP1 can directly induce activation of STAT3 via JAK3 in an EBV-negative NPC line (Figure 2) [65,101]. LMP1 also causes STAT3 phosphorylation at S727 through Protein kinase C-δ (PKCδ) and Extracellular signal-regulated kinase (ERK) in epithelial cells [102,103].

### 4.2. CSMs and Chronic Inflammation during KSHV Latency

Several KSHV oncogenes, including vFLIP, vIL6, vGPCR, K15 and K1, expressed after successful KSHV de novo infection are critical for KSHV oncogenesis. While vGPCR, K15 and K1 are induced during KSHV lytic replication, they are constantly expressed at a low level during KSHV latency [104,105,106]. vGPCR, K15 and K1 are membrane proteins which are partly exposed to the cell surface and chronically activate NF-κB [106,107,108]. While vFLIP is mostly localized in the cytosolic compartment of PEL cells, it initiates NF-κB activation in membrane lipid rafts of PEL cells. Our recent findings suggest that Cell Adhesion Molecule 1 (CADM1), which is highly induced during KSHV de novo infection, is essential for vFLIP to maintain chronic NF-κB activation in PEL cells [71]. We also found that vGPCR, which is a close homolog to mammalian chemokine receptors CXCR1 and CXCR2, required CADM1 to activate NF-κB (Figure 2) [71]. A recent report suggests the possible role of CADM1 in the regulation of STAT3 activation in squamous cell carcinoma progression [109]. vGPCR also chronically activates STAT3 phosphorylation at Y705 [110]. However, the mechanism of STAT3 activation by vGPCR is poorly understood. In addition, whether CADM1 is required for vGPCR to activate STAT3 is not known. Studies have shown that chronic activation of STAT3 by KSHV vIL6 is critical for the survival of PEL cells. The survival and proliferation of PEL cells were significantly diminished after knocking down CADM1 [111,112]. It is possible that CADM1 may also be required for vIL6-mediated STAT3 activation in PEL cells.

### 4.3. CSMs and Chronic Inflammation during HTLV-1 Latency

The two HTLV-1 oncogenes, Tax and HBZ play important roles in maintaining chronic inflammation to promote HTLV-1 pathogenesis [113,114]. Tax expression during the early stages of HTLV-1 latent infection is crucial for chronic NF-κB activation and induction of inflammation that is necessary for the development of HTLV-1-associated diseases. When Tax is conjugated by K63-linked polyubiquitination in membrane lipid rafts, it mediates the persistent activation of NF-κB by triggering chronic phosphorylation and activation of the IKK kinase complex [115]. This activation of IKK leads to the phosphorylation and proteasomal degradation of IκBα. Recent findings suggest that Tax interacts with CADM1 in membrane lipid rafts and initiates IKK kinase complex activation. Tax also interacts with the CSM Scribble, a prominent regulator of T-cell polarity, to mediate activation of NF-κB in T-cells [116]. Tax also mediates STAT3 activation by inducing interleukin-6 receptor (IL-6R) expression in HTLV-1-infected cells [117]. When Tax expression is lost in about 60% of ATLL patients HBZ protein and RNA promote the growth and survival of the leukemic cells [118,119]. While HBZ inhibits the classical NF-κB pathway probably to facilitate escape from host immune responses and avoid senescence [120], it maintains chronic inflammation in mice similar to those in HTLV-1-infected individuals [121]. HBZ interacts with STAT3 and modulates the IL-10/JAK/STAT signaling pathway to promote T cell proliferation [122]. Although CADM1 is one of the most highly expressed molecules in ATLL cells, it is not clear whether it is essential for HBZ-induced inflammation in ATLL cells.

## 5. Role of CSMs in the Negative Regulation of NF-κB and STAT3 Inhibition 

Mounting evidence has revealed that transient induction of inflammation by NF-κB and STAT3 in response to injury or infection is essential to activate immune responses. Chronic inflammation mediated via NF-κB and STAT3 is inhibited by NF-κB negative regulatory proteins (A20 and Cyld) and STAT3 negative regulatory (SOCSs) proteins to prevent chronic inflammation-induced autoimmune diseases and cancers and to restore homeostasis and tissue repair. Most of these NF-κB and STAT3 negative regulators are induced by NF-κB and STAT3 themselves and function in negative feedback loops. A20 and Cyld are deubiquitinating enzymes that inhibit the ubiquitination and activation of key IKK kinase complex activators, including TRAFs and RIP1, in receptor-mediated NF-κB activation [123,124,125]. The DUB activity of A20 is dispensable for the inhibition of IKK and NF-κB activation as it can inhibit NF-κB activation in a non-catalytic manner via zinc-finger 4 (ZF4) and ZF7 [126,127]. In addition, A20 disrupts E2:E3 ubiquitin enzyme complexes independently of DUB activity to inhibit NF-κB signaling [128]. Interestingly, it is not well understood how these molecules are activated in response to NF-κB and STAT3 activation. In addition, it is unclear if the CSMs that are involved in the activation of NF-κB and STAT3 initiate the activation of NF-κB and STAT3 negative regulators. Recently, we and others have shown that the HTLV-1 Tax protein targets NF-κB negative regulatory complexes to maintain chronic activation of NF-κB [29,125,128,129,130]. Interestingly, the CSM CADM1 is critical for Tax to target NF-κB negative regulatory complexes to maintain chronic activation of NF-κB in HTLV-1-infected cells [115] (Figure 2). CADM1 is also critical for the KSHV oncogenes vGPCR and vFLIP to maintain chronic activation of NF-κB in KSHV-infected PEL cells. Thus, it is possible that CADM1 may also inhibit NF-κB negative regulatory complexes. Because CADM1 is highly upregulated in both HTLV-1 and KSHV-infected cells, where STAT3 is chronically activated, it is likely that CADM1 may also be involved in directly or indirectly inhibiting STAT3 negative regulatory mechanisms. It would not be surprising if CADM1 is also induced by EBV infection and involved in chronic activation of both NF-κB and STAT3 by either targeting NF-κB and STAT3 inhibitory mechanisms or directly activating NF-κB and STAT3 signaling. 

## 6. Conclusions

A large number of CSMs are involved in maintaining cell–cell communication and conveying various signals from the extracellular environment to inside the cell nucleus in multi-cellular organisms. While conveying extracellular signals, they also recognize various pathogens present in the extracellular environment and transmit danger signals. However, certain pathogenic viruses such as EBV, KSHV and HTLV-1 have evolved multiple strategies to dysregulate the normal function of CSMs to benefit viral replication or persistence. In this review, we have highlighted select CSMs that are involved in facilitating the entry of EBV, KSHV and HTLV-1 into their target cells, NF-κB and STAT3 activation, and inflammation essential for EBV, KSHV and HTLV-1 pathogenesis. Chronic inflammation mediated by CSMs during de novo infection by these pathogens creates a favorable environment for the initiation of latency and subsequent diseases. Although studies have shown that acute induction of inflammation by NF-κB and STAT3 in response to infection eliminates or limits pathogen spread, chronic inflammation driven by NF-κB and STAT3 activation create favorable environments for EBV, KSHV and HTLV-1. Oncogenes expressed during EBV, KSHV or HTLV-1 latency further dysregulate the function of NF-κB and STAT3 negative regulators to maintain chronic activation of NF-κB and STAT3. Most of these viral oncogenes hijack host CSMs to dysregulate NF-κB and STAT3 negative regulators in lymphotropic virus-infected cells. Therefore, research into the precise mechanisms of CSMs-mediated chronic inflammation in EBV, KSHV and HTLV-1 infections may result in new therapies to target the cancers caused by these viruses.

## Figures and Tables

**Figure 1 biology-09-00390-f001:**
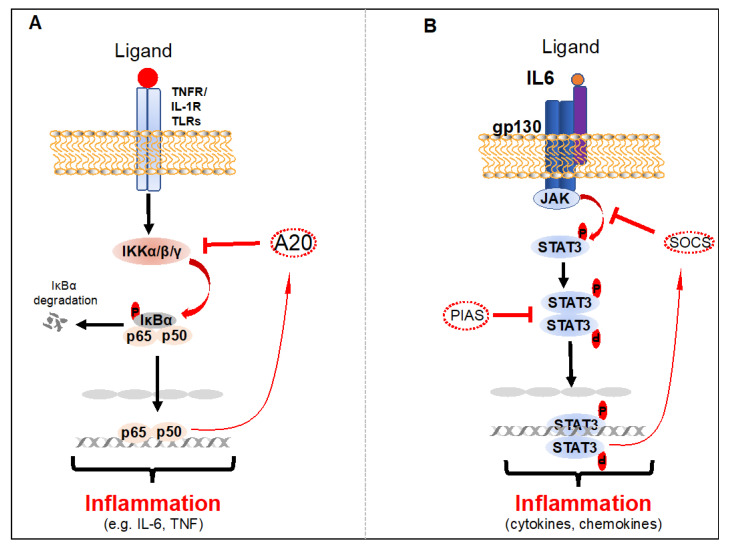
Activation and negative regulation of NF-κB and STAT3. (**A**) Activation of NF-κB in response to the stimulation of TNFR1, IL-1R, or TLRs. A20 is induced by NF-κB in the TNFR1, IL-1R, or TLRs pathways, functioning as a negative feedback loop. (**B**) Activation and negative regulation of the STAT3 signaling pathway. The activation of STAT3 is mediated in response to cell surface receptor (growth factor or cytokine receptors) stimulation that leads to the recruitment and activation of the JAK family of kinases, which, in turn, phosphorylate STAT3. Phosphorylated STAT3 dimerize and translocate to the nucleus, where they directly regulate gene expression. This activation loop can be turned off by PIAS and SOCS proteins.

**Figure 2 biology-09-00390-f002:**
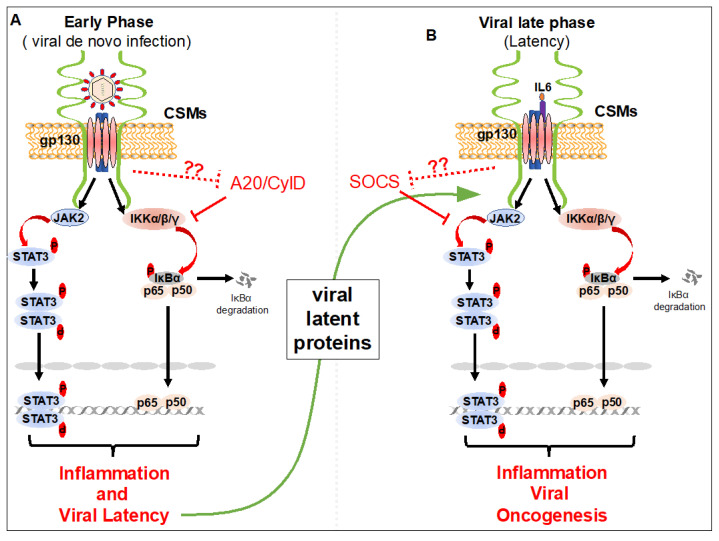
Cell surface molecules-induced STAT3 and NF-κB activation during the early and late phases of infection by lymphotropic viruses. (**A**) Upon viral (Epstein Barr virus (EBV) or Κaposi’s sarcoma herpesvirus (KSHV)) attachment to cell surface molecules, STAT3 and NF-κB activation, viral entry into target cells, and viral latency establishment occurs. Rapid inflammation induced by STAT3 and NF-κB activation helps to establish viral latency. (**B**) During viral latency (viral late phase) the viral latent proteins and/or oncogenes, vIL6 and vFLIP of KSHV, LMP1 and BLIF1 of EBV, and Tax and HBZ of human T-cell leukemia virus type 1 (HTLV-1), are expressed. Viral proteins interact with cell surface molecules and maintain chronic activation of STAT3 and NF-κB, essential for EBV, KSHV and HTLV-1 oncogenesis. Viral oncogenes may hijack cell CSMs to inhibit the function of A20 and CylD proteins in NF-κB, and SOCS proteins in STAT3 pathways to maintain chronic activation of NF-κB and STAT3 in lymphotropic virus-infected cells.

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
