# Peer review of "Lymphotropic Viruses: Chronic Inflammation and Induction of Cancers"

_biology, 2020, doi:10.3390/biology9110390_

Round 1

Reviewer 1 Report

Harhaj et al. reviewed the crosstalk between leukemia development and dysregulation of CSMs-mediated signaling pathway via virus infection. Overall, the review is well-written, especially the explanation of inflammatory cytokines and their signaling. However, it does not adequately explain the latest clinical findings, including therapeutic agents, and the following points should be added and corrected.

Major:

1, This article is about the "leukemia virus", not the "oncogenic virus". An appropriate title should be considered and changed.

2, Chapter 3 and 4 detail intracellular signals in the inflammatory response, and in recent years there have been a number of reports on HTLV-1, EBV and KSHV infections and the induction of inflammation via innate immunity such as RIG-I and NOD signaling. The mechanisms of innate immunity and subsequent carcinogenesis due to viral infections should not be ignored and should be addressed in this section.

3, In Session 6, the treatments for each of the viral infections are described, but clinical initials, including the latest treatment results, need to be described. In some cases, especially with KSHV drugs, treatment with interferon may be effective. Furthermore, mogamulizumab has been used in the treatment of HTLV-1, which should be explained and the efficacy of the treatment should be described.

Minor:

1, In Figure 1 and in the regend, PIAS is listed, but in line 113 is PIAS3. It should be consistent with the correct term.

2, What is the molecular mechanisms of IKK suppression by A20, and are there important the deubiquitination activities and zinc finger motifs of A20?

Author Response

Reviewer_1: Comments and Suggestions for Authors

Harhaj et al. reviewed the crosstalk between leukemia development and dysregulation of CSMs-mediated signaling pathway via virus infection. Overall, the review is well-written, especially the explanation of inflammatory cytokines and their signaling. However, it does not adequately explain the latest clinical findings, including therapeutic agents, and the following points should be added and corrected.

Major:

Suggestion 1, This article is about the "leukemia virus", not the "oncogenic virus". An appropriate title should be considered and changed.

Answer: We have changed the title to “Lymphotropic viruses: chronic inflammation and induction of cancers”.

Suggestion 2, Chapter 3 and 4 detail intracellular signals in the inflammatory response, and in recent years there have been a number of reports on HTLV-1, EBV and KSHV infections and the induction of inflammation via innate immunity such as RIG-I and NOD signaling. The mechanisms of innate immunity and subsequent carcinogenesis due to viral infections should not be ignored and should be addressed in this section.

Answer: We thank the reviewer for raising this concern. Although it is true that activation of RIG-I and NOD signaling induces inflammation, they also induce anti-viral responses which are antagonized by viral proteins such as Tax of HTLV-1, ORF64 and ORF52 of KSHV, and LMP1, BLRF2, and BPLF1 of EBV. Thus, we focused on possible mechanisms used by these viruses and their oncogenic proteins to induce and maintain chronic inflammation required for HTLV-1, KSHV and EBV oncogenesis

Suggestion 3, In Session 6, the treatments for each of the viral infections are described, but clinical initials, including the latest treatment results, need to be described. In some cases, especially with KSHV drugs, treatment with interferon may be effective. Furthermore, mogamulizumab has been used in the treatment of HTLV-1, which should be explained and the efficacy of the treatment should be described.

 Answer: We have deleted the paragraph on treatments based on another reviewer’s suggestion.

Minor:

1, In Figure 1 and in the legend, PIAS is listed, but in line 113 is PIAS3. It should be consistent with the correct term.

Answer: We have changed PIAS3 to PIAS.

2, What is the molecular mechanisms of IKK suppression by A20, and are there important the deubiquitination activities and zinc finger motifs of A20?

Answer: We have previously demonstrated that A20 disrupts E2:E3 ubiquitin enzyme complexes independently of DUB activity to inhibit NF-kB signaling (Shembade et al. Science 2010 327: 1135). A report from the lab of Dr. James Chen also reported a noncatalytic mechanism whereby A20 binds to NEMO via zinc-finger 7 (ZF7), and blocks IKK activation by TAK1 (Skaug et al. Mol. Cell 2011 44: 559). Therefore, it seems that the DUB activity of A20 is dispensable for NF-kB inhibition, and ZF7 plays an important role.

Reviewer 2 Report

The authors propose a review focused on the role of cell surface molecules (CSMs) in inflammation induced by EBV, KSHV, and HTLV-1 oncogenic viruses infection. Their review is sufficiently clear and up-to-date. Particularly, this review represents a good description of what is known, at molecular level,  in inflammation-related signaling (in particular in the NF-kB-related pathways), occurring during the above mentioned virus infection.  Most of the important points are covered and duly referenced. Actually, the work of authors was facilitated by the fact that they directly contributed to research in this field, with some papers.

However, in the opinion of this reviewer some changes could improve the readability of the paper.

First, the paragraph on treatments of EBV/KSHV/HTLV-1-associated malignancies seems out of the specific purpose of this review and addressed in a too much simple manner: it should be entirely omitted.

Second, the title is a little bit misleading. This review is focused only on three oncogenic viruses, not on all oncogenic viruses. The adjective “chronic” seems not adequate to describe inflammation induced by de novo infection, i.e. by the content of three entire paragraphs. "Chronic" should be better omitted. Finally, the role of CSMs seems not actually the “core” subject of the review. This review seems more focused on inflammatory signaling originated rather on the originating molecules that are only one of the aspects. Thus, authors should try to rephrase the title according to these comments.

Minor point: line 45 , “to develop cancers they are associated with” sounds better than “these cancers”.       

Author Response

Reviewer_2 Comments and Suggestions for Authors

The authors propose a review focused on the role of cell surface molecules (CSMs) in inflammation induced by EBV, KSHV, and HTLV-1 oncogenic viruses infection. Their review is sufficiently clear and up-to-date. Particularly, this review represents a good description of what is known, at molecular level,  in inflammation-related signaling (in particular in the NF-kB-related pathways), occurring during the above mentioned virus infection.  Most of the important points are covered and duly referenced. Actually, the work of authors was facilitated by the fact that they directly contributed to research in this field, with some papers.

However, in the opinion of this reviewer some changes could improve the readability of the paper.

Suggestion 1. First, the paragraph on treatments of EBV/KSHV/HTLV-1-associated malignancies seems out of the specific purpose of this review and addressed in a too much simple manner: it should be entirely omitted.

Answer: Thank you for this suggestion. We have deleted the paragraph on treatments of EBV/KSHV/HTLV-1-associated malignancies.

Suggestion 2. the title is a little bit misleading. This review is focused only on three oncogenic viruses, not on all oncogenic viruses. The adjective “chronic” seems not adequate to describe inflammation induced by de novo infection, i.e. by the content of three entire paragraphs. "Chronic" should be better omitted. Finally, the role of CSMs seems not actually the “core” subject of the review. This review seems more focused on inflammatory signaling originated rather on the originating molecules that are only one of the aspects. Thus, authors should try to rephrase the title according to these comments.

Answer: We have changed the title to “Lymphotropic viruses-induced inflammation and cancers”.

Minor point: line 45 , “to develop cancers they are associated with” sounds better than “these cancers”. 

Answer:”. (we don’t see this in the text)

Reviewer 3 Report

Comments to Author

   The authors describe molecular basis of oncogenic virus-induced chronic inflammation during de novo infection as well as latent infection conditions. The content of the manuscript is solid and attractive in terms of future innovation in cancer treatment. The molecular mechanisms of virus-induced inflammation have been concisely summarized with rational insight.

Major point:

  1. In Fig. 2, activation of STAT3 and NF-κB during infection by oncogenic viruses is shown. However, considering that different viruses including EBV, KSHV, and HTLV-1 may use different signaling pathways to promote persistent activation of STAT3 or NF-κB pathways, Fig. 2 should be depicted separately for each of EBV, KSHV, and HTLV-1, in order to make the figures more effective to show their similarity. Also, given the important role for CADM1 in inflammation induction by KSHV and HTLV-I, I would recommend the authors to add the roles of CADM1 in Fig. 2.

Minor points:

  1. In page 1, the authors describe that “Infection with lymphotropic viruses, including Epstein Barr virus/human herpesvirus 4 (EBV/HH4), Κaposi's sarcoma herpesvirus/human herpesvirus 8 (KSHV/HHV8) and human T-cell leukemia virus type 1 (HTLV-1) is associated with the development of approximately 12-15% cancers worldwide (1)”. According to reference 1, prevalence of lymphotropic viruses-related cancers is not so high. Please verify that the percentage is correct.

  1. In page 4, line 30, the authors describe that “At the same time binding of EBV envelope gp350 to these receptor complexes rapidly induceactivation of IKK and NF-κB (Fig. 2) (60, 61)”. The references 60 and 61 may not support the description. Please include appropriate citations.

Author Response

Reviewer_3 Comments and Suggestions for Authors

The authors describe molecular basis of oncogenic virus-induced chronic inflammation during de novo infection as well as latent infection conditions. The content of the manuscript is solid and attractive in terms of future innovation in cancer treatment. The molecular mechanisms of virus-induced inflammation have been concisely summarized with rational insight.

Major point:

  1. In Fig. 2, activation of STAT3 and NF-κB during infection by oncogenic viruses is shown. However, considering that different viruses including EBV, KSHV, and HTLV-1 may use different signaling pathways to promote persistent activation of STAT3 or NF-κB pathways, Fig. 2 should be depicted separately for each of EBV, KSHV, and HTLV-1, in order to make the figures more effective to show their similarity. Also, given the important role for CADM1 in inflammation induction by KSHV and HTLV-I, I would recommend the authors to add the roles of CADM1 in Fig. 2.

Answer: The key kinases involved in the activation of STAT3 or NF-κB are not different for either EBV, KSHV, or HTLV-1-infected cells (during de novo or latency). Thus, for the sake of simplicity, we have depicted the main and common steps of STAT3 and NF-κB activation during early and late phases of infections by these lymphotropic viruses.

Minor points:

  1. In page 1, the authors describe that “Infection with lymphotropic viruses, including Epstein Barr virus/human herpesvirus 4 (EBV/HH4), Κaposi's sarcoma herpesvirus/human herpesvirus 8 (KSHV/HHV8) and human T-cell leukemia virus type 1 (HTLV-1) is associated with the development of approximately 12-15% cancers worldwide (1)”. According to reference 1, prevalence of lymphotropic viruses-related cancers is not so high. Please verify that the percentage is correct.

 Answer: Thank you for noticing this error. We have deleted the statement that 12-15% of cancers are caused by EBV, KSHV and HTLV-1.

  1. In page 4, line 30, the authors describe that “At the same time binding of EBV envelope gp350 to these receptor complexes rapidly induce activation of IKK and NF-κB (Fig. 2) (60, 61)”. The references 60 and 61 may not support the description. Please include appropriate citations.

      Answer: We have included more appropriate references to support this statement.

Round 2

Reviewer 1 Report

The revised version of the paper is better than the previous one. However, if I were to be greedy, the section on the latest clinical findings and future developments in therapeutics would be better understood by the reader if it was elaborated on rather than deleted. However, even without these, the review is generally good, as the pathogenic mechanisms in the leukemia virus are well explained.